# Diazinon residues levels in farm-gate *Brassica oleracea* var. *acephala* of Kimira-Oluch smallholder farm improvement project, Kenya

**George Odoyo Oromo**⬥*[1], **Philip Okinda Owuor**[1], **Bowa Kwach**[1], **Peter Otieno**[2]

**1** Department of Chemistry, Maseno University, Maseno, Kenya, **2** Pest Control Products Board, Kisumu, Kenya

\* oroge2002@gmail.com

## Abstract

Diazinon insecticide, though associated with human health impacts, is popularly used in the production of *Brassica oleracea* var. *acephala* (kale) at the Kimira-Oluch Smallholder Farmers Improvement Project (KOSFIP), Kenya. Diazinon controls insect pests that lower quality and profitability of produce. The preharvest interval of diazinon in kale is 12 days which may not be observed by farmers with inadequate appreciation of Good Agricultural Practices (GAP). Since the extent of GAPs adoption at KOSFIP has not been evaluated, it remains unclear whether diazinon residues levels in kale of KOSFIP could be a health risk to the consumers. Diazinon residues levels and corresponding health risks in farm-gate kale at KOSFIP were determined. Cross-sectional survey based on snowball sampling identified 40 farms applying diazinon on the vegetable. Triplicate samples were collected from each farm for residue analysis, using the QuEChERS method, and LC-ESI-MS/MS analysis. Standard normal distribution function f(z) revealed ≈78% of farm-gate samples had detectable residual diazinon levels and 70% were above the Codex MRL of 0.05 mg/kg. Continued application of diazinon on kale at KOSFIP is exposing consumers to short-term health risks. Efforts must be intensified to ensure GAP are adopted. The estimated farm-gate samples with health risk indices for children and adults (HRIc and $HRI_A$) >1.0 were 64% and 26%, respectively. The residual levels are therefore potential health risks to both children and adults. Farm-gate residual levels and resultant partial HRI were comparatively higher than findings of most previous studies. Inappropriate label PHI and malpractices against GAP may be responsible for high residual levels. There should be regular surveillance and trainings of farmers on GAP for sustainable production of kale in the Lake Victoria region. Use of diazinon on kale should be discouraged and intensive routine pesticide residue screening be enhanced for conventional vegetable produce.

**Data availability statement:** All relevant data are within the paper and its supporting information files.

**Funding:** The author(s) received no specific funding for this work.

**Competing interests:** The authors have declared that no competing interests exist.

## Introduction

Pesticides enhance crop production for the increasing global population towards the realization of sustainable development goals (SDGs) of the United Nations [1]. However, synthetic pesticides residues have become ubiquitous contaminants in the environment [2,3]. The residues pose serious to lethal health hazards to non-target organisms through inhalation, contact and ingestion of contaminated foodstuffs [1]. Approximately 30% (based on mass) of human food is of vegetable origin, mostly consumed raw or semi-processed. Vegetables can therefore be sources of pesticide residues to human beings more than other food groups [4] since most vegetable production uses various pesticides. Ingestion of contaminated foodstuffs is a major exposure route to pesticide residues [5]. For sustainable vegetable production, good agricultural practices (GAPs) and pesticide screening of produce is essential. However, in most low-and middle-income countries (LMICs), evaluation of pesticide residues in vegetables treated with pesticides during production are minimal.

*Brassica oleracea* var*. acephala* (Kale) is grown in many parts of the world [6–8] as food and for its numerous health benefiting metabolites that minimize the risk of degenerative diseases like cancer [9,10]. Pests and pest-related diseases are the major hindrances to quality and quantity of produce [11]. The survey of vegetable farmers reported multiple pests in kale production. Aphids, Diamondback moth, and cutworms and cabbage loopers were reported by 97%, 75%, and 60% of farmers, respectively [12]. Other pests including caterpillars, flea beetles, whiteflies, bugs, thrips, and webworms have also been reported in all agroecological zones but in varying prevalences [13]. With increasing climate change impacts, insect pests have been reported to increase rapidly [14]. Consequently, cultivation of vegetables, including kale involves frequent application of broad spectrum pesticides for the management of pests that attack the roots and foliage [13,15]. However, techniques of application, especially in LMICs are less often guided by GAPs and consumers may be exposed to chronic residue levels.

In Kenya, diazinon (*O,O*–diethyl–*O*-(2-isopropyl-6-methyl-4-pyrimidinyl) phosphorothioate) is one of the broad-spectrum insecticides registered for use in *kale* production [16]. The contact organophosphate insecticide transforms in *vivo* into its oxon forms (diazoxon and hydroxydiazoxon) which inhibits acetylcholinesterase enzyme (AChE) in insect pests with no hydrolytic activity [17]. AChE inhibition results in excessive accumulation of acetylcholine (ACh) and overstimulation of cholinergic nervous system [18]. Physiological functions in the insect pests are paralysed and the intoxicated insect pest dies. Subsequently, aphids, thrips, red spider mites, whiteflies, Diamondback moth, fruit flies, fruit worms, locusts and grasshoppers have been controlled using diazinon. Diazinon on kale leaves is expected to undergo dissipation through chemical degradation processes and other forms of physical transformations including surface wash-off. The total dissipation effect including efficacy on insect pests should reduce diazinon residues to levels below the Codex maximum residue limit (MRL) of 0.05 mg/Kg when diazinon is applied according GAPs [19].

Kimira-Oluch Smallholder Farmers Improvement Project (KOSFIP) is an irrigation project located in Homa Bay County in the Republic of Kenya. While the area is

characterized by rapid fluctuations in weather parameters, it consistently experiences hot and humid climatic conditions with scanty rainfall of high variability in duration and amounts [20,21]. Studies carried out in regions with similar varying weather parameters, especially relative humidity and temperatures, reported high build-up of aphids, Diamondback moth, caterpillars, beetles and painted bugs [22–24]. At KOSFIP, the smallholder farmers of kale manage the insect pests by using the affordable and accessible diazinon. However, diazinon application conditions on kale recommends a pre-harvest interval (PHI) of 12 days [16] which may not be observed by farmers with inadequate appreciation of GAPs [2]. Consequently, it is possible that residues of diazinon in *kale* grown in KOSFIP may be above Codex MRL. Unfortunately, screening of vegetable produce for pesticide residue safety compliance has never been done at KOSFIP and the diazinon residues in farm gate *kale* vegetables might be a health risk to the consumers. This study examined whether the levels of residual diazinon in farm gate kales of KOSFIP differ significantly from the tolerable Codex MRL of 0.05 mg/Kg. Consequently, a survey of diazinon residue levels in farm gate vegetables and the health risks the residue levels may pose were investigated. The survey results of farm gate diazinon residues in *kale* vegetables are reported herein.

## Materials and methods

### Study area

This study was carried out within the Kimira-Oluch Smallholder Farmers Improvement Project (KOSFIP) site. KOSFIP is an irrigation project located in Rachuonyo (Kimira site) and Homa Bay (Oluch site) sub-counties of Homa Bay County in Kenya (Fig 1) [25]. The site lies between latitudes 0° 20' S and 0° 30' S and longitudes 34° 30' E and 34° 39' E at an altitude of 1154 m above mean sea level along the shores of Lake Victoria [26].

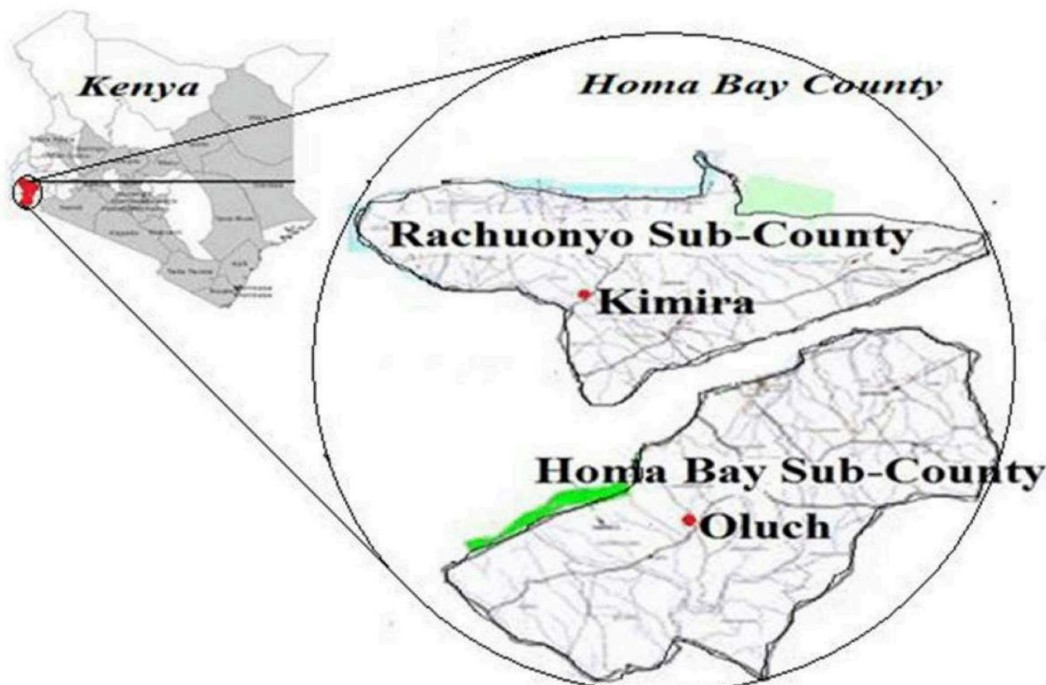

**Fig 1. Map showing the position of KOSFIP inside the map of Kenya with a clear separation of Kimira and Oluch sub-divisions of KOSFIP within Homabay County, Kenya.** Reprinted from [25] under a CC BY licence, with permission from Makone, S. M original copyright (2020).

Kimira site has an area of 1,790 ha out of which 808 ha have been developed into 44 farming blocks whilst Oluch site has an area of 1,308 ha with only 666 ha split into 53 farming blocks have been irrigated [21]. The area is sub-humid with mean annual rainfall of between 740 and 1200 mm with short and long rainy seasons during April-May and November-December, respectively, and mean annual maximum temperatures of $31^0C$ and minimum of $18^0C$ [20,21]. The relative humidity is significantly high ranging between 60 and 75% with potential evapotranspiration rate at 1800 mm and 2000 mm per annum. Apart from the rains, the farms are irrigated, thereby producing vegetables throughout the year. The sites have fertile alluvial soils originating from the nutrient alluvial deposits washed downstream from the rivers and erosions from the Gusii highlands.

### Research design

A cross-sectional survey design based on purposive and snowball sampling techniques were used to identify forty-five farms of *kale* that use diazinon in vegetable production. The survey was carried out during dry season (February to March) in both Kimira and Oluch sites of the project. During the period, Kimira site had 18 active blocks with *kale* while Oluch had 35 sites. In addition, during sampling, Kimira site had 12 farms while Oluch had 33 farms. The criteria for snowballing included same vegetable variety treated with diazinon only at first harvest. Among the forty-five farms, fourty were selected [27]. Kimira had 11 farms while Oluch had 29 farms. The farms were spread equitably to represent Kimira and Oluch sections of the project, considering all the 97 blocks making the project area. Sampling was biased to farms of *kale* that had been treated with diazinon before the first harvest after planting. From every farm, 1 Kg of freshly harvested *kale*, replicated three times were collected to make 120 samples.

### Sample processing, preparation, extraction and partitioning

The processing, preparations, extraction and partitioning of kale vegetable leaf samples for diazinon residues analysis were carried out using Quick Easy Cheap Effective Rugged and Safe (QuEChERS) multi-residue method [28] as adopted and validated by the Analytical Chemistry Laboratories of Kenya Plant Health Inspectorate Services (KEPHIS) method M0326. The validated method modified and optimized extraction of specific vegetable samples, including kale to omit sample clean-up since the vegetable has very low chlorophyl and oil content. The validated modification reported cost effective and a more efficient procedure that eliminates positive and negative false equipment responses.

### Sample processing and preparation

The vegetable samples were coarsely cut with a knife then chopped and homogenized with a Hobart food processor. About 100 g of the homogenized samples were placed in sample containers and were then stored frozen at $-18^oC \pm 5^oC$ in readiness for extraction.

### Extraction and partitioning of samples

A $10.0 \pm 0.1$ g of the homogenous wet samples were weighed into 50 ml centrifuge tubes. Using an automatic pipette, 50 µl of procedural internal standards (dimethoate D6 (10 ppm) and malathion D10 (10 ppm)) were added to the contents of the centrifuge tubes. To the contents of the centrifuge tube, $10.0 \pm 0.2$ ml extraction solvent acetonitrile (MeCN) HPLC grade was added. The tube was immediately closed and shaken vigorously by Geno grinder for 1 minute at 1000 revolutions per minute (rpm). The resulting homogenous mixture in the centrifuge tube was then subjected to liquid-liquid partitioning step using 6.5 g of premixed extraction salts. The extraction salts comprised $4.0 \pm 0.2$ g magnesium sulphate anhydrous for removal of water and salting out MeCN; $1.0 \pm 0.05$ g sodium chloride to increase selectivity of analyte by reducing amount of co-extracted matrix; $1.0 \pm 0.05$ g trisodium citrate dihydrate and $0.5 \pm 0.03$ g disodium hydrogen citrate sesquihydrate as a citrate buffer for pH adjustment. The tube was closed and immediately shaken vigorously by hand to avoid caking. The mixture was again shaken by Geno grinder for 1 minute with 1000 rpm then centrifuged for 5 minutes at 3700 rpm.

An aliquot of 500 μl of the mixture was transferred into a 2.0 ml vial followed with 495 μl of HPLC grade water and 5 μl of injection internal standard dimethoate D6 (10 ppm). The mixture was vortexed to mix for LC-MS/MS analysis. The extracts were directly subjected to quantitative analysis by LC-MS/MS (dMRM) mode.

## Preparation of calibration solutions for method validation

Calibration solutions were prepared using a control matrix containing no detectable residues of diazinon analytes. The control samples were fortified with increasing concentrations of diazinon standard solutions to achieve linearity for LC-MS/MS analysis. Using an automatic pipette, 4 ml of control blank was put into a 15 ml centrifuge tube followed with 4 ml of HPLC grade water and vortexed to mix. Reference standard solutions of diazinon pesticide stocked by KEPHIS were prepared for analysis at concentrations of 0.005 μg/ml, 0.01 μg/ml, 0.02 μg/ml, 0.05 μg/ml, 0.075 μg/ml, 0.1 μg/ml and 0.2 μg/ml for validation of method, and 0.005 μg/ml, 0.02 μg/ml, 0.05 μg/ml, and 0.2 μg/ml for routine analysis using the blank control.

## Instrumentation and instrument specifications

The extracted samples of *kale* were analysed using Liquid Chromatography Quadruple Agilent 6430 LC-MS/MS with standard electron spray ionization (ESI). The HPLC column used was C-18 with an internal diameter of 1.8 μm and a length of 50 cm. The optimization parameters, including solvent gradient, precursor and product ions, and retention times were set as outlined in method M0326. The column temperature was set at 40.0°C. The auto-sampler injection and ejection speed was 200μL/min with an injection volume of 3.00 μL.

The parameters were optimized using a binary mobile phase comprising HPLC grade water (A) and HPLC grade acetonitrile (B), both treated with 0.01% formic acid for enhanced ionization of the analyte molecule. The column timetable was: 0 to 2 minutes (A 95%, B 5%); the next 5 minutes (A and B, 50% each); and the last 11 minutes, (A 5% B 95%).

## Quality control

The 2018 Standard Operating Principles (SOP) number M0326 for QuEChERS Multi-Residue Method for Analysis of Pesticides Residues [28] in high water matrices validated by Analytical Chemistry Laboratory of KEPHIS was used in the determination of residue levels. The quality parameters included repeatability, linearity, accuracy of recovery, method's limits of detection (LOD) and quantitation (LOQ). Calibration curve (Fig 2) was drawn according to analyte ranges of concentration and response to the LC-ESI-MS/MS. This was achieved by using five replicates of different concentrations diluted with blank extract samples. Evaluations of accuracy and precision parameters were done by recovery experiments (recovery range of 93–123%) in which each analyte standard were spiked with blank *kale* slurry in six replicates. The replicates were prepared separately at three different concentrations of 10, 20 and 50 μg Kg$^{-1}$.

Limits of detection (LODs) of each analyte was validated by comparing the signal-to-noise (S/N) ratio magnitude to the background noise obtained from blank sample in the six replicates that presented mean coefficient of variations (CV) of less than 20%. The time window for the signal - to - noise (S/N) ratio was set at t < 2 minutes. LOD was calculated using the mathematical expression [29]:

$$LOD = \frac{(3.9 \times SD\ residuals)}{(Slope\ of\ the\ calibration\ curve)}$$

(1)

Where SD residuals are the standard deviation of residuals

The limits of quantification (LOQs), defined as the minimum concentration of an analyte that can be identified and quantified with 99% confidence, was calculated using the mathematical expression [29]:

$$LOQ = 3.3 \times LOD$$

(2)

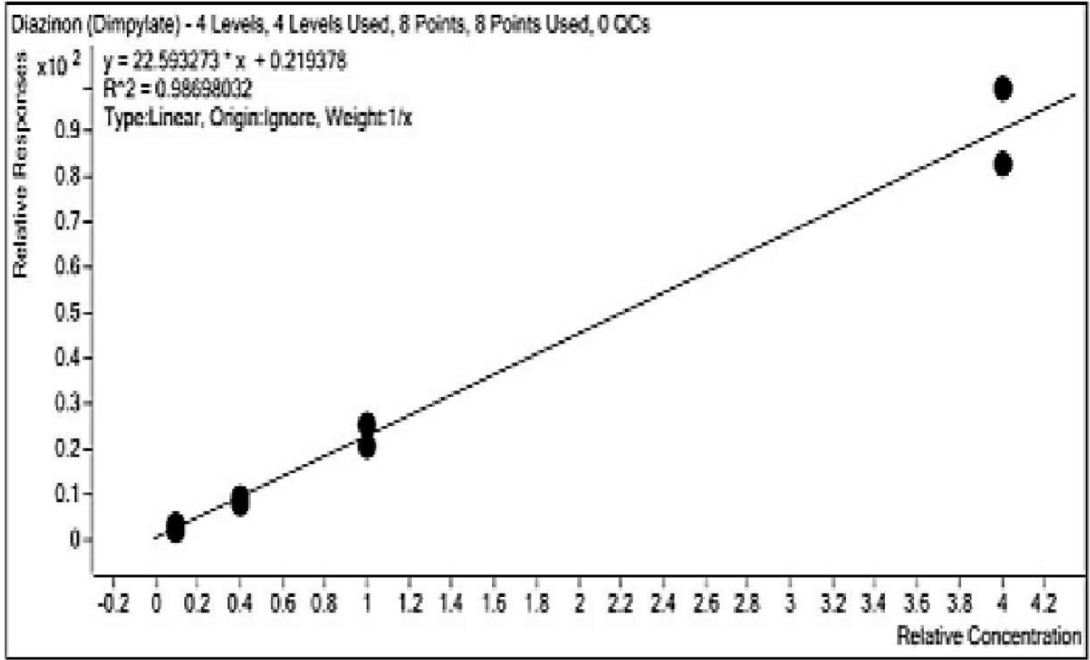

**Fig 2. Calibration curve drawn from varying concentrations of diazinon standards.**

## Analytical determination of residual diazinon

For quantitative analysis of the analytes, 3 µl of the solvent, matrix control, calibration standards and spikes, and samples were injected into the LC-MS/MS instrument. Responses were recorded for both internal standards and samples. A calibration curve of responses against concentration of calibration standards was obtained. The results of concentrations of residues of diazinon for all the samples were calculated from responses obtained from the calibration curve. Respective chromatograms and graphics for quantitation and confirmation were obtained.

## Statistical analysis

The cross-sectional survey data of diazinon residues in farm gate samples were subjected to descriptive statistics for purposes of illustrating measures of central tendency and dispersion: mean concentrations, mode, median, quartiles, standard deviations, minimum and maximum values, range of values and the coefficient of variations (CV). Outliers were checked and tested using univariate techniques (interquartile range method) and the calculations returned no minimum and realistic maximum values. The diazinon residues levels were also compared with Codex MRL values of 0.05 mg/kg and evaluated for health risk assessment. For both residue levels and health risk indices, the standard normal distribution function f(z) was used to determine the proportion of values that lie below and above the tolerable values of MRL and HRI. The values were computed using the formula:

$$Standard\ normal\ variable\ (z) = \frac{(x - \mu)}{\sigma}$$

(3)

Where z is the standard proportion value; x is the acceptable value; µ is the mean of the data set; and, σ is the standard deviation of the data set.

## Residual Pesticide Health Risk Assessment

The kale (single-item) partial residual pesticide health risk indices (pHRI) estimations for children ($HRI_C$) and for adults ($HRI_A$) based on the farm gate samples were estimated using the European Union formula [30]:

$$pHRI = \frac{pEDI}{ADI}$$

(4)

Where, pHRI is the partial health risk index for kale vegetable; pEDI is the estimated *partial* daily intake exposure of diazinon pesticide; and ADI is the acceptable daily intake of the pesticide.

The EU formula provides that the total EDI be determined by multiplying the sample residual pesticide concentration (mg/kg) by the estimated WHO food consumption rate (kg/day), and dividing by the number of the estimated WHO average body weight [31]. pHRI > 1.0 were considered as posing health risks, hence not safe for human health. HRI ≤ 1.0 were considered not posing immediate health risks and thus safe for human health [32,33]. The average daily vegetable intake for an adult of average weight 60 Kg was considered to be 0.345 Kg/person/day while children average daily intake was considered to be 0.232 kg/person/day for average body weight of 10 kg [34]. The maximum acceptable daily intake (ADI) was considered to be 0.003 mg/kg body weight while the acute reference dose (ARfD) was 0.03 mg/kg body weight [35].

## Results and discussion

### Levels of residual diazinon in farm gate baskets of kale at KOSFIP area of Homabay for health risk assessment

Levels of diazinon residues in farm-gate *kale* samples at KOSFIP (Table 1) were determined and statistically analyzed using the standard normal distribution function (f(z)). The findings showed that approximately 92% of all the samples had detectable levels of residual diazinon while 8% had non-detectable levels. Similarly, the percentage of farm gate samples with higher residues of diazinon than the acceptable MRL of 0.05 mg/kg was 70%. The measures of central tendency (mean, median and mode) displayed a positively skewed distribution with a coefficient of variation (CV) of 122% (Table 2). The distribution demonstrated that the residues levels of diazinon in farm-gate *kale* were highly variable with a large range. The standard normal distribution function (f(z)) showed that approximately 64% of the samples could pose health risks to children. Similarly, approximately 26% of the samples could pose health risks to adults. The findings indicate that children consuming *kale* from the study area may be more at risk than adults.

## Discussions

Determination of residual diazinon levels in the farm-gate baskets of *kales* from the KOSFIP area of Homa Bay County was done to assess compliance to Codex standards and for health risk assessment. The findings were comparatively higher than levels reported by similar studies in Ghana [36], Nigeria [37] and other parts of Kenya [38,39], that reported trace levels of diazinon residues with less than 10% of the samples being above the Codex MRL. The variability could be as a result of multiple factors associated with inadequate training of farmers on good agricultural practices and poor surveillance by respective national authorities [40]. Consequently, farmers in KOSFIP area require regular surveillance and training on use of diazinon. The findings also suggest that the recommended diazinon application conditions of rates and pre-harvest intervals may be too short and not suitable for the study area, respectively. In addition, the findings suggest that KOSFIP farm-gate vegetables need thorough washing before cooking to reduce the leaf surface residual levels on the vegetables.

Given that food safety standards encourage infinitesimal residual levels of pesticide residues [19], a positively skewed distribution with all measures of dispersion falling below Codex MRL (0.05 mg/kg) would be most preferable. However, the findings of this study demonstrate mean residual diazinon concentrations much higher than Codex MRL. As a result, the high diazinon residues levels in the farm gate samples may pose health risks to consumers. Subsequently, most of

**Table 1. Levels of diazinon residues in farm-gate baskets of *Brassica oleracea* var. *acephala* from selected KOSFIP farms and resultant EDI and HRI for children and adults.**

| Serials of Farms | Mean Conc. (mg/Kg) | pEDI$_C$ | pEDI$_A$ | pHRI$_C$ | pHRI$_A$ |
|---|---|---|---|---|---|
| 1 | 0.03 ± 0.00 | 0.0007 | 0.0002 | 0.2320 | 0.0575 |
| 2 | 0.00 ± 0.00 | 0.0000 | 0.0000 | 0.0000 | 0.0000 |
| 3 | 0.82 ± 0.02 | 0.0190 | 0.0047 | 6.3413 | 1.5717 |
| 4 | 0.05 ± 0.01 | 0.0012 | 0.0003 | 0.3867 | 0.0958 |
| 5 | 1.07 ± 0.05 | 0.0248 | 0.0062 | 8.2747 | 2.0508 |
| 6 | 0.05 ± 0.01 | 0.0012 | 0.0003 | 0.3867 | 0.0958 |
| 7 | 0.05 ± 0.01 | 0.0012 | 0.0003 | 0.3867 | 0.0958 |
| 8 | 0.02 ± 0.00 | 0.0005 | 0.0001 | 0.1547 | 0.0383 |
| 9 | 0.00 ± 0.00 | 0.0000 | 0.0000 | 0.0000 | 0.0000 |
| 10 | 0.06 ± 0.01 | 0.0014 | 0.0003 | 0.4640 | 0.1150 |
| 11 | 0.83 ± 0.04 | 0.0193 | 0.0048 | 6.4187 | 1.5908 |
| 12 | 0.09 ± 0.00 | 0.0021 | 0.0005 | 0.6960 | 0.1725 |
| 13 | 0.05 ± 0.01 | 0.0012 | 0.0003 | 0.3867 | 0.0958 |
| 14 | 0.06 ± 0.01 | 0.0014 | 0.0003 | 0.4640 | 0.1150 |
| 15 | 0.15 ± 0.03 | 0.0035 | 0.0009 | 1.1600 | 0.2875 |
| 16 | 0.40 ± 0.01 | 0.0093 | 0.0023 | 3.0933 | 0.7667 |
| 17 | 0.03 ± 0.00 | 0.0007 | 0.0002 | 0.2320 | 0.0575 |
| 18 | 0.02 ± 0.00 | 0.0005 | 0.0001 | 0.1547 | 0.0383 |
| 19 | 0.03 ± 0.01 | 0.0007 | 0.0002 | 0.2320 | 0.0575 |
| 20 | 0.65 ± 0.04 | 0.0151 | 0.0037 | 5.0267 | 1.2458 |
| 21 | 0.95 ± 0.04 | 0.0220 | 0.0055 | 7.3467 | 1.8208 |
| 22 | 0.08 ± 0.00 | 0.0019 | 0.0005 | 0.6187 | 0.1533 |
| 23 | 0.01 ± 0.00 | 0.0002 | 0.0001 | 0.0773 | 0.0192 |
| 24 | 0.09 ± 0.00 | 0.0021 | 0.0005 | 0.6960 | 0.1725 |
| 25 | 0.04 ± 0.00 | 0.0009 | 0.0002 | 0.3093 | 0.0767 |
| 26 | 0.20 ± 0.02 | 0.0046 | 0.0012 | 1.5467 | 0.3833 |
| 27 | 0.54 ± 0.04 | 0.0125 | 0.0031 | 4.1760 | 1.0350 |
| 28 | 0.00 ± 0.00 | 0.0000 | 0.0000 | 0.0000 | 0.0000 |
| 29 | 0.05 ± 0.00 | 0.0012 | 0.0003 | 0.3867 | 0.0958 |
| 30 | 0.12 ± 0.02 | 0.0028 | 0.0007 | 0.9280 | 0.2300 |
| 31 | 0.04 ± 0.00 | 0.0009 | 0.0002 | 0.3093 | 0.0767 |
| 32 | 0.65 ± 0.04 | 0.0151 | 0.0037 | 5.0267 | 1.2458 |
| 33 | 0.44 ± 0.03 | 0.0102 | 0.0025 | 3.4027 | 0.8433 |
| 34 | 1.06 ± 0.03 | 0.0246 | 0.0061 | 8.1973 | 2.0317 |
| 35 | 0.06 ± 0.04 | 0.0014 | 0.0003 | 0.4640 | 0.1150 |
| 36 | 0.74 ± 0.04 | 0.0172 | 0.0043 | 5.7227 | 1.4183 |
| 37 | 1.00 ± 0.17 | 0.0232 | 0.0058 | 7.7333 | 1.9167 |
| 38 | 0.04 ± 0.00 | 0.0009 | 0.0002 | 0.3093 | 0.0767 |
| 39 | 0.41 ± 0.01 | 0.0095 | 0.0024 | 3.1707 | 0.7858 |
| 40 | 0.62 ± 0.05 | 0.0144 | 0.0036 | 4.7947 | 1.1883 |

Replicates per farm = 3; EpDI$_A$ – Expected *partial* Daily Intake for adults; EpDI$_C$ – Expected *partial* Daily Intake for children; pHRI$_A$ – *partial* Health Risk Index for adults; pHRI$_C$ – *partial* Health Risk Index for children; LOD - Limits of Detection (0.0100 mg/Kg); 7.5% ≤ LOD ≤ 92.5%; MRL ≥ 70%

**Table 2.** Measures of central tendency and dispersion for levels of diazinon residues in farm-gate baskets of *Brassica oleracea* var. *acephala* from KOSFIP area and resultant EDI and HRI for children and adults.

| Measure of Central Tendency & Dispersion | Mean Conc. (mg/Kg) | $EDI_C$ | $EDI_A$ | $HRI_C$ | $HRI_A$ |
|---|---|---|---|---|---|
| Mode | 0.050 | 0.001 | 0.000 | 0.387 | 0.096 |
| Minimum | 0.000 | 0.000 | 0.000 | 0.000 | 0.000 |
| 1st Quartile | 0.040 | 0.001 | 0.000 | 0.309 | 0.077 |
| Median | 0.060 | 0.001 | 0.000 | 0.464 | 0.115 |
| 3rd Quartile | 0.560 | 0.013 | 0.003 | 4.331 | 1.073 |
| Interquartile range | 0.520 | 0.012 | 0.003 | 4.021 | 0.997 |
| Maximum | 1.070 | 0.025 | 0.006 | 8.275 | 2.051 |
| Range | 1.070 | 0.025 | 0.006 | 8.275 | 2.051 |
| Maximum Outlier | 1.340 | 0.031 | 0.008 | 10.363 | 2.568 |
| Mean | 0.290 | 0.007 | 0.002 | 2.243 | 0.556 |
| S. Deviation | 0.354 | 0.008 | 0.002 | 2.738 | 0.679 |
| CV (%) | 122.100 | 122.100 | 122.100 | 122.100 | 122.100 |

CV – Coefficient of Variation; $EDI_A$ – Expected Daily Intake for adults; $EDI_C$ – Expected Daily Intake for children; $HRI_A$ – Health Risk Index for adults; $HRI_C$ – Health Risk Index for children; Mean concentration of 0.00 represents concentrations below Limits of Detection (LOD). Since all values fall below the calculated maximum outlier value, all data were used in the interpretation.

the farm gate *kale* vegetables treated with diazinon at KOSFIP may not therefore be safe for human consumption. These findings were similar to the levels reported in some vegetables of Bangladesh [41,42], Kuwait [43,44], Nigeria [45], Ghana [46] and Sudan [47], where over 30% of samples reported residue levels above the Codex MRLs. The estimated partial daily intake (pEDIC) and resulting health risk indices for children (HRIC) indicate that children consuming the vegetable from the study area have higher chances of developing diazinon related health problems [48,49]. Cumulative exposures, when additional diazinon intake occurs may pose much higher health risks to both children and adults. Given that data on residual levels with computed EDI and HRI for children are not available, comparisons of health risk factors for children in different regions have not been made.

On the other hand, the estimated partial daily intake for adults ($EDI_A$) and resultant health risk index for adults ($HRI_A$) were comparatively lower than ratios for children (Table 1 and Table 2). The findings were similar to residual levels of diazinon in cauliflower of Bangladesh [50], tomatoes of Spain [51] and Iran [52], respectively. The effect of the residual levels of diazinon on EDI and HRI for adults were higher than the findings in Chinese kale [53], spring onion, parsley onion and ginger vegetables in Thailand [54]. In addition, the results were also higher than those reported for yard long bean in Bangladesh [50,55], apple in Pakistan [56], *T. occidentalis* and *C.argentea* in Nigeria [37], *kale* [38,39] and tomatoes [38] in Kenya.

However, the results and the resultant EDI and HRI were lower than ratios reported on eggplant and tomatoes in Pakistan [41,57], watermelon [45], spinach and onions [58] in Nigeria, tomatoes in Ghana [36] (36), cucumber and tomatoes in Sudan [47]. Given the high variability displayed (coefficient of variation (C.V.) of 122%), the farmers and consumers of the vegetables are likely to be exposed to diazinon associated health risks [43,44]. These health challenges may threaten human population in the study area. On the same note, though the adults have a lower partial mean ratio of 0.556 versus 2.243 for children, cumulative exposure and continuous consumption of this popular vegetable may cumulatively raise the exposure levels and result in devastating health impacts [59].

The presence of inappropriate levels of diazinon residues in the leaves of *kale* may be a consequence of non-adherence to good agricultural practices (GAPs) such as failing to observe the application conditions of dosage and pre-harvest intervals. Such disregard are popular with smallholder farmers in developing countries for locally

consumed vegetables [2,60,61] and in regions where surveillance and monitoring activities are inadequate. The unacceptable residue levels could be due to the inability of the farmers to interpret the rates as prescribed on the labels or utter disregard of the rates, and inadequate surveillance of farmers on the use of pesticides. Inappropriate levels may also be attributed to inability of the farmers in the study area to consider other viable methods of pest control provided by Integrated Pest Management (IPM) guidelines [43,62] even during heavy infestations. To manage the unacceptable residue levels and related health problems, training of farmers on GAP and alternatives to chemical pest control should be initiated in the study area. National regulatory agencies should equally strengthen surveillance on farmers with emphasis on observance of pre-harvest intervals and pesticide dosages. The public should also be sensitized on the need to reduce the risks through proper culinary processing of the vegetables before consuming them. Lastly, the inappropriate residue levels may result from label application conditions which were extrapolated from field trials done in other regions but not suitable for the study area. It is therefore well justified to constantly monitor residual levels of diazinon in the vegetable and to develop pesticide safety level monitoring policies to mitigate resultant health problems due to unacceptable residue levels.

## Conclusions

In Kenya, Brassica *oleracea* var. *acephala* is grown by over 90% of smallholder farmers for subsistence. The vegetable is affordable and popular in most food outlets. Contaminants including pesticide residues may pose health risks to significant populations. The findings in this study showed that most of the farm-gate kale vegetables treated with diazinon within the KOSFIP area of Homa Bay County of Kenya had higher than tolerable residue levels. The residual diazinon quantities may pose significant health risks to the consumers. The high levels suggested that most farmers of *kale* who use diazinon within the KOSFIP area may not be observing good agricultural practices (GAPs). Consequently, there is need to restrict the use of diazinon on *kale* at KOSFIP and to consider alternative pesticides with shorter PHIs to reduce high levels of residual diazinon in farm gate vegetables. In addition, there is need for a survey study into the extent of good agricultural practices (GAPs) with respect to pesticide use in the production of *kale* and other vegetables within the KOSFIP area of Homa Bay County. On the same note, there is need for cumulative health risk index (HRI) determination for various age groups with estimated local weights, consumption rates and pesticide safety levels (PSL) adapted to the local environment. Finally, there is need for a comparative study on the effects of processing (washing, blanching, cooking) to establish if such processing methods significantly reduces diazinon residue levels in kale of KOSFIP area.

This study has provided baseline information required for the establishment of Good Agricultural Practices (GAP) towards use of diazinon in production of *kale* vegetables in the KOSFIP area. The study has provided a basis for discouraging the use of diazinon in the production of *kale* in KOSFIP and other environments.

## Supporting information

**S1 File. Table of laboratory sample codes, analyte molecule, and residual concentration of farm-gate samples from KOSFIP.** XXX.
(PDF)

**S2 File. Qualitative analysis data for internal standards (dimethoate) and calibration standards for diazinon, as well as the resultant calibration curve for determining residual concentrations of analytes.** XXX.
(PDF)

**S2 File. Quality control parameters, including precision, mean recovery, linearity, calibration standards and chromatograms for selected analyte samples.** XXX.
(PDF)

## Acknowledgments

This work was supported by the West Kenya Union Conference of Seventh-day Adventists bursary fund. The cost of analysis was supported by a subsidy and waiver from the Kenya Plant Health Inspectorate Services (KEPHIS) at the Analytical Chemistry Laboratories, Karen - Nairobi.

## Author contributions

**Conceptualization:** George Odoyo Oromo.

**Data curation:** George Odoyo Oromo.

**Formal analysis:** George Odoyo Oromo, Philip Okinda Owuor.

**Investigation:** George Odoyo Oromo.

**Methodology:** George Odoyo Oromo.

**Project administration:** George Odoyo Oromo.

**Resources:** George Odoyo Oromo.

**Supervision:** Philip Okinda Owuor, Bowa Kwach.

**Validation:** Philip Okinda Owuor, Peter Otieno.

**Visualization:** George Odoyo Oromo.

**Writing – original draft:** George Odoyo Oromo.

**Writing – review & editing:** Philip Okinda Owuor, Bowa Kwach, Peter Otieno.

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
