## [Decision Letter · Decision Letter 0]

8 Jan 2025

PONE-D-24-38603Diazinon residues levels in farm-gate Brassica oleracea var. Acephala of Kimira-Oluch Smallholder Farm Improvement Project, KenyaPLOS ONE

Dear Dr. Oromo,

Thank you for submitting your manuscript to PLOS ONE. After careful consideration, we feel that it has merit but does not fully meet PLOS ONE’s publication criteria as it currently stands. Therefore, we invite you to submit a revised version of the manuscript that addresses the points raised during the review process.

Comments from PLOS Editorial Office: We note that one or more reviewers has recommended that you cite specific previously published works. As always, we recommend that you please review and evaluate the requested works to determine whether they are relevant and should be cited. It is not a requirement to cite these works. We appreciate your attention to this request.

We look forward to receiving your revised manuscript.

Kind regards,

Sanjay Kumar Gupta, Ph.D.

Academic Editor

PLOS ONE

Journal requirements: When submitting your revision, we need you to address these additional requirements.

1. Please ensure that your manuscript meets PLOS ONE's style requirements, including those for file naming. The PLOS ONE style templates can be found at https://journals.plos.org/plosone/s/file?id=wjVg/PLOSOne_formatting_sample_main_body.pdf and https://journals.plos.org/plosone/s/file?id=ba62/PLOSOne_formatting_sample_title_authors_affiliations.pdf.

2. Please amend either the title on the online submission form (via Edit Submission) or the title in the manuscript so that they are identical.

4. We note that your Data Availability Statement is currently as follows: [All relevant data are within the manuscript and its supporting information files]

5. We notice that your figures are uploaded with the file type 'Other'. Please amend the file type to 'Figure'. Please ensure that each Supporting Information file has a legend listed in the manuscript after the references list.

Reviewers' comments:

Reviewer's Responses to Questions

**Comments to the Author**

1. Is the manuscript technically sound, and do the data support the conclusions?

Reviewer #1: Yes

Reviewer #2: No

2. Has the statistical analysis been performed appropriately and rigorously? 

Reviewer #1: Yes

Reviewer #2: No

3. Have the authors made all data underlying the findings in their manuscript fully available?

Reviewer #1: Yes

Reviewer #2: No

4. Is the manuscript presented in an intelligible fashion and written in standard English?

Reviewer #1: Yes

Reviewer #2: Yes

5. Review Comments to the Author

Reviewer #1: General comments:

The article has notable merits; however, the extraction and clean p method did not written properly. Therefore, the article can be accepted for publications after necessary corrections, and clarifications.

Specific comments:

In page 4 ,line 118- 136: Extraction and partitioning of samples should be re-written. In this section the authors only needs to mention the extraction and cleanup of the real samples, it is not wise to mention the procedures of addition of standard of diazinon. The addition of standard or the fortification of standard should be mentioned in the preparation of fortified samples in the method validation part. In this section the authors only mention the extraction of diazinon, however, they did not mention the procedures of clean up. It should be addressed properly?

In page 7, line 172-176: How the authors have measured the mean concentration of Blank sampes in case of LOD and LOQ determination? The determination of LOQ was not appropriate. In case of LOQ determinations please follow the CODEX or EU (SANTE) guidelines.

In page 8. Iine 200-202: In case of HRI estimation, the ADI value is mentioned for specific pesticides in all food items that the consumers have taken in a day. However the EDI was measured for the specific pesticides in a single food items not for all food items that the consumers have taken in a day, so how it is comparable? Please clarify.

In page 12, ine 256: The reference (39) you have cited was an old one, please replace with the new one: Nahar KM., Khan MSI., Habib M., Hossain SM., Prodhan MDH and Islam MA. Health risk assessment of pesticide residues in vegetables collected from northern part of Bangladesh. Food Research 4 (6), 2281-2288

In page 13, ine 280: The reference (46) you have cited was an old one, please replace with the new one: Habib M., A. Kaium, M.S.I. Khan, M. D. H. Prodhan, N. Begum, M. T. I. Chowdhury, M.A. Islam. 2021. Residue level and health risk assessment of organophosphorus pesticides in eggplant and cauliflower collected from Dhaka city, Bangladesh. Food Research 5 (3), 369-377

The authors are requested to add very recent references in introduction and discussions chapter, you may add the following articles: 1. Multiple pesticide residue determination in major vegetables purchased from Gazipur district of Bangladesh.

2. Analysis of organophosphorus pesticide residues in selected vegetables purchased from Narsingdi district of Bangladesh using QuEChERS Extraction.

Reviewer #2: General comments:

The article has no merits at this stage. The article has not been written with proper scientific questions and hypotheses. The article required substantial improvement before acceptance on the given concerns below.

Specific comments:

Abstract:

It should be concise with quantified data and more information about Diazinon and its uses in Brassica.

The research implications are very wide. Please give the precise implications of the study.

Introduction:

The sentences “These conditions promote the rapid spread of vegetable pests and diseases (20,21). The smallholder farmers manage the pests by use of synthetic pesticides.” are very broad and irrelevant to study. Please give specific mentions of insect pests and control by which insects in Brassica field in particular locations recommended by the authority.

No scientific hypothesis and research gaps are provided by the authors in the introduction part.

“Brassica oleracea var. acephala” once it has been used; after that, it gives a short name for this.

Result section:

Separate result and discussion part

Line 235-236: “The data set (Table 1) had no outliers”. What was the method of outlier detection?

Calibration, validation and peak of pesticide detection are to be provided for review purposes.

Specific recommendations and utility of the present research at the end of the discussion and conclusion are to be provided in the revised manuscript.

6. PLOS authors have the option to publish the peer review history of their article (what does this mean? ). If published, this will include your full peer review and any attached files.

**Do you want your identity to be public for this peer review?** For information about this choice, including consent withdrawal, please see our Privacy Policy .

Reviewer #1: **Yes: ** Mohammad Dalower Hossain Prodhan

Reviewer #2: **Yes: ** Dr. Jaipal Singh Choudhary

---

## [Author Response · Author response to Decision Letter 1]

2 Mar 2025

We have made effort to respond to all reviewer and editor comments and we believe that the revised manuscript is much better because of your guidance.

---

## [Editor Report · Decision Letter 1]

22 Apr 2025

Diazinon residues levels in farm-gate Brassica oleracea var. Acephala of Kimira-Oluch Smallholder Farm Improvement Project, Kenya

PONE-D-24-38603R1

Dear Dr. George Odoyo Oromo

We’re pleased to inform you that your manuscript has been judged scientifically suitable for publication and will be formally accepted for publication once it meets all outstanding technical requirements.

Kind regards,

Sanjay Kumar Gupta, Ph.D.

Academic Editor

PLOS ONE
---

## [Editor Report · Acceptance letter]

PONE-D-24-38603R1

PLOS ONE

Dear Dr. Oromo,

I'm pleased to inform you that your manuscript has been deemed suitable for publication in PLOS ONE. Congratulations! Your manuscript is now being handed over to our production team.

Kind regards,

on behalf of

Dr. Sanjay Kumar Gupta

Academic Editor

PLOS ONE